# Mutual Information Based Bayesian Graph Neural Network for Few-shot Learning

**Kaiyu Song**[1,2]    **Kun Yue**[1,2]    **Liang Duan**[1,2]    **Mingze Yang**[1,2]    **Angsheng Li**[2,3]

[1]Key Lab of Intelligent Systems and Computing of Yunnan Province, Yunnan University, Kunming, China
[2]School of Information Science and Engineering, Yunnan University, Kunming, China
[3]School of Computer Science and Engineering, Beihang University, Beijing, China

## Abstract

In the deep neural network based few-shot learning, the limited training data may make the neural network extract ineffective features, which leads to inaccurate results. By Bayesian graph neural network (BGNN), the probability distributions on hidden layers imply useful features, and the few-shot learning could improved by establishing the correlation among features. Thus, in this paper, we incorporate mutual information (MI) into BGNN to describe the correlation, and propose an innovative framework by adopting the Bayesian network with continuous variables (BNCV) for effective calculation of MI. First, we build the BNCV simultaneously when calculating the probability distributions of features from the Dropout in hidden layers of BGNN. Then, we approximate the MI values efficiently by probabilistic inferences over BNCV. Finally, we give the correlation based loss function and training algorithm of our BGNN model. Experimental results show that our MI based BGNN framework is effective for few-shot learning and outperforms some state-of-the-art competitors by large margins on accuracy.

## 1 INTRODUCTION

Few-shot learning aims to learn novel concepts from only one or a few annotated samples, which is an interesting problem and has received a lot of attention recently [Ma et al., 2020]. Different from traditional machine learning models built on a large amount of training data, few-shot learning is defined for scenarios with limited supervised experience [Zheng et al., 2021]. It is challenging to fulfill efficient few-shot learning, since extracting effective and representative features often requires large-scale training datasets [Gairola et al., 2020].

To find more useful features for few-shot learning, several methods have been proposed and one popular solution is Bayesian graph neural network (BGNN) [Jospin et al., 2020], which is a graph neural network (GNN) to describe the uncertain relationships among features in datasets. By using Bayesian approximation over uncertainty, BGNN could extract more effective features to improve the performance of few-shot learning tasks [Hasanzadeh et al., 2020]. However, it is still difficult to achieve a highly accurate few-shot learning method based on BGNN, since the limited training data is insufficient for neural networks to catch the most useful features and makes over-smooth and over-fitting much more intensively.

One advantage of BGNN is that the probability distribution of the features extracted from hidden layers could be used to build a larger feature space. If we could eliminate redundant features and restrain the feature space in terms of the convergence with limited samples, more useful features could be extracted. Note that the correlation of probability distributions among hidden layers implies useful information of features. Thus, by scaling the correlation of feature spaces, we could improve the effectiveness of feature extraction of BGNN for few-shot learning. For example, given two neighbor hidden layers in a BGNN for face recognition, the prior layer extracts *location* and the next layer extracts *shape*. Then, based on the correlation between these two kinds of features, we could make the next layer to extract both *shape* and *location* instead of just *shape*.

Probability distributions reflect both the features in hidden layers and corresponding propagation operations like graph convolution. If the correlation among features could be extracted and described, it could be used to make hidden layers of BGNN share with more information and find more useful features by maximizing the correlation [Kipf and Welling, 2017]. For this purpose, we adopt mutual information (MI) to describe the correlation quantitatively, and formulate the process of MI maximization to make BGNN share as much information as possible by following the forward flow in BGNN training [Gabrié et al., 2018].

*Accepted for the 38^{th} Conference on Uncertainty in Artificial Intelligence* (UAI 2022).

However, calculating MI is not trivial even if the probability distribution functions (PDFs) have been given [Amjad et al., 2019]. It is known that Bayesian network (BN) is a famous framework for uncertain knowledge representation and inference via a directed acyclic graph (DAG) of random variables with conditional probability parameters [Koller and Friedman, 2009]. Thus, we use BN with continuous variables (BNCV) [Li and Mahadevan, 2018] to effectively approximate MI.

In this paper, we propose an innovative framework to establish the correlation among features in hidden layers of BGNN to improve the accuracy of few-shot learning tasks. Specifically, we first build our framework based on a Bayesian graph convolution neural network with adaptive connection sampling (BGS) [Hasanzadeh et al., 2020] (an efficient version of BGNN). We then approximate the probability distributions of features extracted from hidden layers by the relaxed Bernoulli distribution in BGS. Thus, the continuity of the probability distributions in BNCV could be guaranteed. We then define the nodes of BNCV as the marginal distributions extracted from features, and establish the correlation between two neighbor layers and connect the end of the prior pair with the start of the next pair based on the forward flow when training BGS. Thus, the DAG of BNCV could be constituted. Although the distributions of the nodes in a BNCV cannot be directly obtained, the Bayesian approximation of the relaxed Bernoulli distributions based on Dropout has already provided the conditional probability function (CPF) for BNCV. Consequently, the approximation of MI could be fulfilled by using CPF and Monto Carlo integration based on the probabilistic inferences over BNCV. Finally, we provide the correlation based loss function and training algorithm of our BGNN model.

Our main contributions are summarized as follows:

- We propose an innovative framework to extract effective features for few-shot learning by establishing the correlation among features from the probability distributions in BGS.

- We build BNCV efficiently from the Dropout in hidden layers of BGS and approximate the MI values effectively based on the probabilistic inferences over BNCV to describe the correlation quantitatively.

- We provide the loss function by incorporating with the MI-based correlation and propose the training algorithm of our BGNN model.

- We conduct extensive experiments on Cora and Citeseer datasets, and the results show that our proposed framework is effective for few-shot learning and outperforms some state-of-the-art competitors by large margins on accuracy.

## 2 RELATED WORK

Few-shot learning aims to learn novel concepts from only one or a few examples, which is an interesting and challenging problem in practical applications [Gao et al., 2021]. Recently, many meta-learning [Zhang et al., 2018] and transfer learning [Wang et al., 2020] methods have been proposed to solve this problem. Most of these methods are combined with a variety of deep learning models, where the correlation among features is usefully provided. For example, the correlation could be obtained by using GNN to propagate structural information [Garcia and Bruna, 2018], and a similarity metric is established to achieve the correlation between two similar samples for few-shot image segmentation [Gairola et al., 2020].

However, describing correlation as the hidden feature for few-shot learning is still challenging. Thus, several methods have been proposed to establish the concept of correlation. [Yan et al., 2019] give the concept of correlation between global and local features based on the dual attention network. [Gao et al., 2020] conceptualize the correlation between existing and new relations via embedding. [Yao et al., 2020] provide a method to learn the correlation from auxiliary graphs via knowledge transfer. To evaluate the correlation explicitly, MI-based methods have been adopted. [Di et al., 2020] use MI to build a 2-depth adjacent matrix to leverage the correlation, and [Wan et al., 2020] use MI to enrich the representation of knowledge extracted from correlation. Moreover, Graphical Mutual Information (GMI) has been proposed to measure the correlation between input graphs and high-level hidden representations [Peng et al., 2020] based on graph embedding. By these methods, the correlation could be evaluated, but still cannot be calculated quantitatively even MI is adopted.

Note that MI could not be easily calculated and usually approximated in practice [Gabrié et al., 2018]. Thus, the generative adversarial neural network based method is proposed to approximate the CPF of density and marginal distributions ultimately [Abbasnejad et al., 2019]. However, these methods are often inefficient [Gabrié et al., 2018]. To solve this issue, the neural network based conditional MI was proposed as the approximation of the MI [Mukherjee et al., 2019], but it does not hold in BGNN for few-shot learning due to the limited training data.

Integrating the deep learning and Bayesian model is the subject with much attention to make interpretation for neural networks or infer the conditional (or even causal) relations and corresponding uncertainty. For example, [Rohekar et al., 2018] propose the method for unsupervised structure learning of deep neural networks by casting the problem of neural network structure learning as a problem of BN structure learning. [Krishnan et al., 2017] propose a unified algorithm to efficiently learn a compiled inference network and the generative model simultaneously for non-linear state

space models to mimic the posterior distribution. [Wang and Yeung, 2020] survey the models and applications of Bayesian deep learning to tightly integrate deep learning and Bayesian models for establishing a comprehensive artificial intelligence system with the capabilities of perception and probabilistic inferences. Different from these methods, we adopt BN for evaluating the correlation efficiently, and build BNCV based on the probability distributions of hidden layers in BGS.

# 3   PROBLEM FORMULATION

First, we formulate some concepts as the basis of later discussion.

A dataset with limited training data is represented as $D(G, F, Y)$, where $G$ is the undirected graph, $F$ is the set of original features, and $Y$ is the set of labels.

Taking as input the training dataset $D$, a BGS contains $I$, $O$, $L$, and $B$, where $I$ (i.e., $I = D$ for the convenience of expression) and $O$ is the input and output respectively, $L$ is the set of graph convolution operations and $B$ is the set of relaxed Bernoulli distributions of all hidden layers, where

- $La(G)$ is the function of passing structural information, such as Laplace decomposition [Kipf and Welling, 2017] for a given undirected graph $G$, given the activation function $\sigma(\cdot)$.
- $L = \{L_1, \ldots, L_n\}$, $B = \{B_1, \ldots, B_n\}$, $L_0 = I$, $L_i = \sigma(La(G)L_{i-1})B_i (1 \leq i \leq n)$.
- $O = Softmax(L_n)$ , and $n$ is the depth (i.e., number of layers) of BGS.

A BNCV contains $V$ and $E$, where $V$ is the set of nodes (i.e., random variables), $E$ is the set of directed edges, where there is a set of conditional probabilities to quantify the dependencies among the nodes in $V$. The DAG of BNCV is represented as $G^d(V, E)$. To build a BNCV from BGS, we consider generating $V$ based on $B$ and $E$ from the forward flow between neighbor layers in $L$.

MI can be formulated by entropy [Di et al., 2020] w.r.t. the probability distribution of $(L_{i-1}, L_i)$ as follows

$$MI(P(L_{i-1}), P(L_i)) = H(P(L_i)) - H(P(L_i|L_{i-1}))$$
(1)

where $P(\cdot)$ is the PDF of random variables and $H(\cdot)$ is the entropy of $P(\cdot)$.

The correlation between a pair of neighbor layers is defined as $(L_{i-1}, L_i)$ and obtained by the approximation of MI, denoted as $\widetilde{\mathcal{M}}(L_{i-1}, L_i)$. Correspondingly, the correlation of the whole framework is defined as $\mathcal{M}\{(L_1, L_2), \ldots, (L_{n-1}, L_n)\}$, abbreviated as $\mathcal{M}(L)$.

We cast our problem of correlation evaluation as the problem of calculating MI-based correlation, which could be

efficiently implemented by the probabilistic inferences over BNCV. Meanwhile, we include correlation into the loss function for training our model, which also makes BGS to possibly make use of the correlation among hidden layers.

# 4   METHODOLOGY

## 4.1   FRAMEWORK

First, we propose the innovative framework, BGS based on MI (BGSMI), which consists of a BGS and a BNCV, shown as Figure 1. Then, we describe the ideas of BGS, BNCV, MI approximation and training of BNCV, respectively.

### 4.1.1   BGS

To represent the correlation based on limited training data, $(L_{i-1}, L_i)$ could be established directly by following the forward flow between two neighbor layers of BGS.

To establish $\{(L_1, L_2), \ldots, (L_{n-1}, L_n)\}$, we limit the propagation scope of correlation, stated in Theorem 1, since only the prior layer influences the result of the current layer according to the formulation of the graph convolution operation $L_i = \sigma(La(G)L_{i-1})B_i$.

**Theorem 1.** *The correlation w.r.t. BGS is between every pair of neighbor layers $L_{i-1}$ and $L_i$ $(1 < i \leq n)$.*

By the graph convolution operation, the Dropout in $B$ could be used to approximate the CPF between every pair of neighbor layers, denoted as $P(L_i|L_{i-1}, La(G)W_{i-1})$, where $W_{i-1}$ is the set of parameters of $L_{i-1}$. $O = Softmax(L_n)$ is adopted as $e^{Z_j}/\sum_{j=1}^{|L_n|} e^{Z_j}$ to generate the output, where $Z_j$ is the $j$th feature generated by $L_n$.

### 4.1.2   BNCV

In a BNCV, the DAG represents the correlation between two neighbor layers in BGS and CPF is the conditional probability functions for each node in BNCV. To build a BNCV, we consider the following two aspects.

First, the marginal distribution $MP(B_i)$ w.r.t. $L_i$ is equal to $P(B_i)$, which is extracted from $L_i$ and regarded as the node in BNCV. Thus, we generate $V$ of BNCV w.r.t. $B$ of BGS. Following the forward flow in BGS, $V_i$ w.r.t. $B_i$ has the relationship with $V_{i-1}$ w.r.t. $B_{i-1}$, since the correlation reflects the dependence relationship in BNCV by a directed edge. $E$ can be generated and further $G^d(V, E)$ can be built by following the forward flow between neighbor layers.

Second, according to Theorem 1, the CPF of each node is defined as

$$P(L_i|L_{i-1}) = P(L_i|L_{i-1}, La(G)W_{i-1})$$
(2)

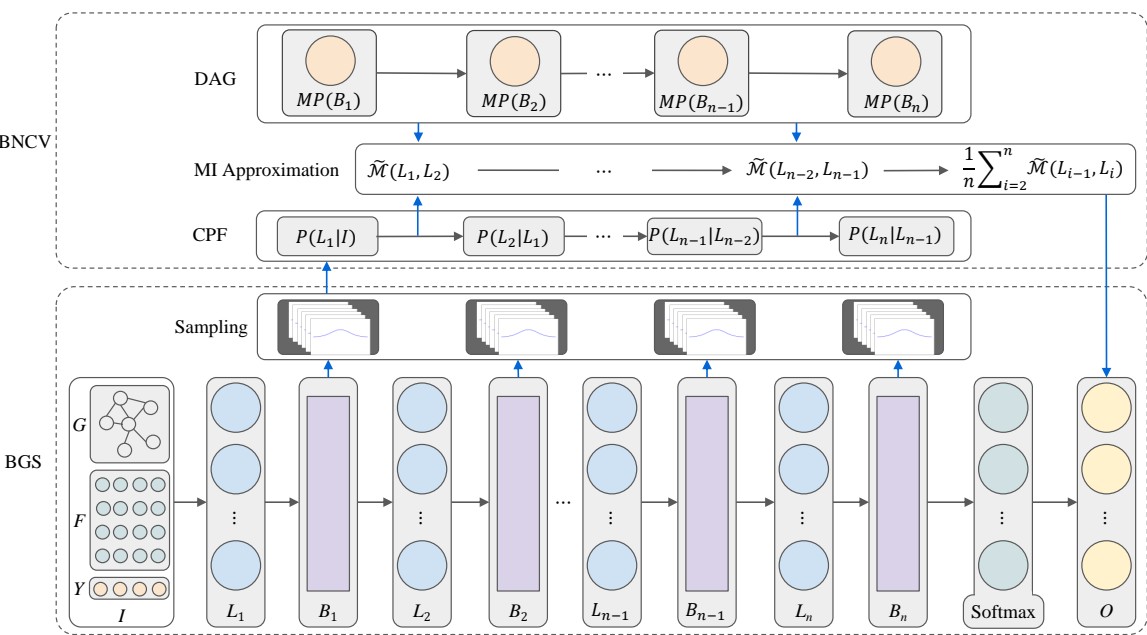

Figure 1: The framework of BGSMI. The network structure of BGS is shown at the bottom. Built from BGS, the BNCV with DAG and CPFs are shown at the top. Blue arrows among Sampling, CPF, MI approximation and DAG show the approximation of $P(L_i|L_{i-1})$ by probabilistic inferences over BNCV. Black arrows show the direction of forward flow between neighbor hidden layers in BGS.

Note that Equation (2) describes the conditional distribution between two nodes in BNCV and $P(L_i|L_{i-1})$ is analogous to the conditional probability table (CPT) in BN with discrete variables. In terms of the structure of the BGSMI, $P(L_i|L_{i-1})$ could be calculated as

$$P(L_i|L_{i-1}) \approx Bernoulli(\pi_i) \approx B_i(a_i, b_i) \qquad (3)$$

where $Bernoulli(\pi_i)$ is the Bernoulli distribution with parameter $\pi_i$, and $B_i(a_i, b_i)$ is the relaxed Bernoulli distribution w.r.t. $L_i$ with parameters $a_i$ and $b_i$.

Thus, we could calculate $P(L_i|L_{i-1})$ for each pair $(L_{i-1}, L_i)$ $(1 < i \leq n)$ over BNCV.

## 4.2 BNCV BASED MI APPROXIMATION

We use MI to describe the correlation between $L_{i-1}$ and $L_i$, denoted as $\widetilde{\mathcal{M}}(L_{i-1}, L_i)$ and approximated as follows

$$\widetilde{\mathcal{M}}(L_{i-1}, L_i) = H(P(L_i)) + H(P(L_{i-1})) \\ - H(P(L_{i-1}, L_i)) \qquad (4)$$

Note that $P(L_{i-1}, L_i)$ could be calculated by the probabilistic inferences over BNCV, and thus the correlation of the whole framework is denoted as $\mathcal{M}(L)$ and defined as

$$\mathcal{M}(L) = \frac{1}{n} \sum_{i=2}^{n} \widetilde{\mathcal{M}}(L_{i-1}, L_i) \qquad (5)$$

Equation (5) could be approximated during one epoch of BGS training by probabilistic inferences over BNCV. Adopting the lower bound of $H(P(L_{i-1}, L_i))$, we make the following transformation

$$H(P(L_{i-1}, L_i)) \approx max\{H(P(L_{i-1})), H(P(L_i))\}. \qquad (6)$$

Thus, $\widetilde{\mathcal{M}}(L_{i-1}, L_i)$ in Equation (4) could be formulated as

$$\widetilde{\mathcal{M}}(L_{i-1}, L_i) = H(P(L_{i-1})) + H(P(L_i)) - max\{\cdot\}, \qquad (7)$$

where $max\{\cdot\}$ denotes $max\{H(P(L_{i-1})), H(P(L_i))\}$ for the convenience of expression, and the entropy is calculated by

$$H(P(L_i)) = -\int P(L_i) \log_2 P(L_i) \, dx. \qquad (8)$$

By the Monto Carlo integration [Jospin et al., 2020], Equation (8) could be transformed as

$$H(P(L_i)) \approx -t \sum_{Sp(L_i)} \log_2 P(L_i), \qquad (9)$$

where $Sp(\cdot)$ denotes the sample space of variables, $t = \frac{1}{Ns(Sp(L_i))}$, and $Ns(\cdot)$ denotes the number of possible variables in $Sp(\cdot)$.

Importantly, the sample space is equal to the number of samples during the process of sampling [Jospin et al.,

2020]. Based on the independent relationships represented in BNCV, $P(L_i)$ could be formulated as

$$P(L_i) = \int P(L_i, L_{i-1}, \ldots, L_1) \, d(Sp(L_{i-1}, \ldots, L_1)). \tag{10}$$

By Monto Carlo integration, we have

$$P(L_i) \approx \frac{1}{Ns(sl)} \sum_{sl} \frac{P(L_i, L_{i-1}, \ldots, L_1)}{P(L_{i-1}, \ldots, L_1)}, \tag{11}$$

where $sl$ denotes $Sp(L_{i-1}, \ldots, L_1)$, and $\sum_{sl} \frac{P(L_i, L_{i-1}, \ldots, L_1)}{P(L_{i-1}, \ldots, L_1)}$ denotes the fraction for all possible combinations of $P(L_i, L_{i-1}, \ldots, L_1)$ and $P(L_{i-1}, \ldots, L_1)$ in $Sp(L_{i-1}, \ldots, L_1)$ for the convenience of expression.

By sampling over the relaxed Bernoulli distributions $B$ in BGSMI, we could obtain $P(L_i|L_{i-1})$ efficiently by Equation (3). It is worth noting that the parameters $a_i$ and $b_i$ of $B_i$ have already been calculated during the training of BGS. Then, using the chain rule, the joint probability distribution of $\{L_i, L_{i-1}, \ldots, L_1\}$ could be obtained as

$$\begin{aligned} P(L_i, L_{i-1}, \ldots, L_1) &\approx \prod_{j=2}^{i} P(L_j|L_{j-1}) \\ &= P(L_i|L_{i-1}) \prod_{j=2}^{i-1} P(L_j|L_{j-1}) \\ &\approx P(L_i|L_{i-1}) P(L_{i-1}, \ldots, L_1). \end{aligned} \tag{12}$$

By Equation (11) and Equation (12), we have

$$H(P(L_i)) \approx \frac{-1}{Ns(Sp(L_i))} \sum_{Sp(L_i)} \log_2 P(L_i|L_{i-1}). \tag{13}$$

Finally, by Equation (7) and Equation (13), $\widetilde{\mathcal{M}}(L_{i-1}, L_i)$ in BGSMI could be appropriated as

$$\begin{aligned} \widetilde{\mathcal{M}}(L_{i-1}, L_i) &= -\frac{1}{Ns(Sp(L_i))} \sum_{Sp(L_i)} \log_2 P(L_i|L_{i-1}) \\ &- \frac{1}{Ns(Sp(L_{i-1}))} \sum_{Sp(L_{i-1})} \log_2 P(L_{i-1}|L_{i-2}) - max\{\cdot\}. \end{aligned} \tag{14}$$

Thus, $\widetilde{\mathcal{M}}(L_{i-1}, L_i)$ could be approximated by the CPFs in BNCV directly. The procedure of the approximation of Equation (5) is given in Algorithm 1, whose complexity is $O(n|W_B|)$.

## 4.3 TRAINING ALGORITHM

Following $(L_1, L_2), \ldots, (L_{n-1}, L_n)$, the correlation between neighbor layers is incorporated into BGS by calculating the correlation of the whole framework. To intensify the

---

**Algorithm 1** Approximation of MI-based Correlation

 **Input:** $I$
 **Parameters:** $W_B$, the weights in $B$
 **Output:** $\mathcal{M}(L)$
1: Let $t$ be an empty list with equal size to $W_B$
2: $i \leftarrow 1$
3: **while** $i \leq |t|$ **do**
4:    Sample $P(L_i|L_{i-1})$ by Equation (3)
5:    Calculate $\widetilde{\mathcal{M}}(L_{i-1}, L_i)$ by Equation (14)
6:    $t[i] \leftarrow \widetilde{\mathcal{M}}(L_{i-1}, L_i))$
7:    $i \leftarrow i + 1$
8: **end while**
9: **return** $\mathcal{M}(L) \leftarrow \frac{1}{|t|} \sum_{i=2}^{n} \widetilde{\mathcal{M}}(L_{i-1}, L_i)$ // Approximation of Equation (5)

---

correlation between neighbor layers in BGS, we maximize Equation (5) in the loss function of BGSMI, since the larger the correlation, the less the uncertainty between neighbor layers. Thus, the loss function of BGSMI is formulated as

$$\mathcal{L} = \mathcal{L}_B + \ln \mathcal{M}(L), \tag{15}$$

where $\mathcal{L}_B$ is the loss function of BGS [Hasanzadeh et al., 2020], formulated as

$$\begin{aligned} \mathcal{L}_B = \mathbb{E}_{q(L,B)} &\ln P(Y|F, L, B) \\ &- KL(q(L,B)||p(L,B)) + \xi \sum_i^{|F|} I_i^2, \end{aligned} \tag{16}$$

where $q(\cdot)$ and $p(\cdot)$ are the distributions of random variables, $KL(\cdot)$ is the Kullback-Leibler (KL) divergence [Jospin et al., 2020], $I_i \in I$, and $\xi$ is a constant.

To avoid under-fitting, we remove $\xi \sum_i^{|F|} I_i^2$ from Equation (16), and thus rebuild the loss function as

$$\begin{aligned} \mathcal{L} = \mathbb{E}_{q(L,B)} &\ln P(Y|F, L, B) \\ &- KL(q(L,B)||p(L,B)) + \ln \mathcal{M}(L). \end{aligned} \tag{17}$$

Therefore, BGSMI could be trained by minimizing the loss function in Equation (17) via gradient descent. The above idea of BGSMI training is summarized in Algorithm 2, whose complexity is $O(n|W_B| + |W_L|T)$.

## 5 EXPERIMENTS

We evaluate BGSMI to answer the following questions:

**Q1:** How does BGSMI perform in terms of the accuracy compared with other state-of-the-art models on limited training data?

**Q2:** How does the MI-based correlation in BGSMI alleviate the over-fitting and over-smooth in few-shot learning?

**Q3:** How does noise impact the accuracy of BGSMI based few-shot learning?

**Algorithm 2** Training BGSMI

    **Input:** $I$
    **Parameters:** $T$, the number of epochs; $lr$, the learning rate
    **Output:** $W_L$, the weights in $L$; $W_B$, the weights in $B$
1:  $i \leftarrow 0$
2:  Randomly initialize $W_L^i$ and $W_B^i$
3:  **while** $i < T$ **do**
4:     Generate $V$ and $E$ from BGS to build $G^d(V, E)$
5:     Calculate $P^c$ based on $G^d(V, E)$, $W_L^i$ and $W_B^i$
6:     Calculate $\mathcal{L}$ by $W_L^i$, $W_B^i$ and $P^c$// By Equation (17)
7:     $(\nabla \mathcal{L}_L, \nabla \mathcal{L}_B) \leftarrow \nabla \mathcal{L}$
8:     $W_L^{i+1} \leftarrow (W_L^i - lr * \nabla \mathcal{L}_L)$
9:     $W_B^{i+1} \leftarrow (W_B^i - lr * \nabla \mathcal{L}_B)$
10:    $i \leftarrow i + 1$
11: **end while**
12: **return** $W_L^T, W_B^T$

## 5.1 EXPERIMENT SETTINGS

**Datasets.** We used two benchmarks of citation networks for graph node classification [Kipf and Welling, 2017], shown in Table 1. Cora records the citation network publication [1] and Citeseer records the citation information of papers released by Citeseer [2].

Table 1: Statistics of datasets.

| Dataset | Nodes | Edges | Features per node | Classes |
|---------|-------|-------|-------------------|---------|
| Cora | 2,078 | 5,429 | 1,433 | 7 |
| Citeseer | 3,327 | 4,732 | 3,073 | 6 |

**Comparison methods.** We carefully chose six state-of-the-art methods as competitors for BGSMI:

- **GCN** (graph convolutional network) [Kipf and Welling, 2017] uses the graph convolution layer with Laplace transform to handle the graphical structure.

- **GAT** (graph attention network) [Veličković et al., 2018] uses attention to simulate Laplace transform and operates on graph-structured data, leveraging masked self-attentional layers based on graph convolutions or their approximations.

- **CHEB** [Defferrard et al., 2016] is a convolutional neural network on graphs with fast localized spectral filtering and applies the fast localized spectral filtering to improve the GCN without concerning the entire graph.

- **GMI** (graphical mutual information with standard structure) [Peng et al., 2020] measures the correlation

between input graphs and high-level hidden representations, and generalizes conventional mutual information computations, concerning node features and graphical structure.

- **BGS** (Bayesian GCN with adaptive connection sampling) [Hasanzadeh et al., 2020] is an efficient version of BGNN.

- **MAML** (model-agnostic meta-learning) [Finn et al., 2017] is a classic meta-learning algorithm to learn the network parameters in deep learning models using a two-step strategy.

- **RALE** (relative and absolute location embedding) [Liu et al., 2021] aligns different tasks toward learning a transferable prior by using the relative and absolute location embedding to solve over-fitting in few-shot node classification on graphs.

**Implementations.** We considered the following two experimental variables: depth of the layers (2, 4, and 6 layers), and inclusion of noise (adding noise in the training set satisfying normal Gaussian distribution). For the Cora dataset, we used 140 nodes, 500 nodes and 1000 nodes as the training set, validation set and test set, respectively. For the Citeseer dataset, we used 120 nodes as the training set, 500 nodes and 1000 nodes as the validation set and test set, respectively. Models of GCN, GAT and CHEB use the pre-training strategy.

**Metric.** We used accuracy to evaluate the effectiveness of BGSMI based few-shot learning, defined as the ratio of the number of correct predictions by the neural network to the number of samples.

**Hyperparameters.** We concerned four hyperparameters and fixed the learning rate to 0.001 for GMI and 0.005 for the other models, the decay rate of L2 regularization to $5e - 3$, and the number of epochs to 400. The kernel sizes of graph convolutions in the comparison methods are the number of features per node multiplied by that of features per node, 128 times of the number of features per node, 512 times of the number of features per node, $512 \times 512$ and $128 \times 128$. The kernel sizes of graph convolutions in BGSMI are 256 times of the number of features per node, $256 \times 256$, $256 \times 128$, $128 \times 64$ and $64 \times 64$.

**Environment.** Our experiments were run on a machine with an Intel i9 3.6GHz CPU, 128GB RAM and RTX3090 GPUs. All codes were written in PyTorch.

## 5.2 EXPERIMENTAL RESULTS

**Accuracy of BGSMI based few-shot learning.** With various sizes of samples in the support and query sets, the accuracy of BGSMI based few-shot learning is compared with that of the comparison methods, reported in Table 2, where '$n$-way $k$-shot' means $n$ samples, included in the support set and query set in each batch, respectively.

---

[1] https://linqs-data.soe.ucsc.edu/public/lbc/cora.tgz

[2] https://csxstatic.ist.psu.edu/downloads/data

Table 2: Accuracy with different sized training sets.

| Method | Cora | | | Citeseer | | |
|--------|------|------|------|----------|------|------|
| | 1-way 3-shot | 1-way 5-shot | 3-way 5-shot | 1-way 3-shot | 1-way 5-shot | 3-way 5-shot |
| GCN | 0.542 | 0.673 | 0.797 | 0.543 | 0.634 | 0.732 |
| GAT | 0.541 | 0.592 | 0.762 | 0.385 | 0.672 | 0.823 |
| CHEB | 0.651 | 0.753 | 0.801 | 0.563 | 0.752 | 0.781 |
| GMI | 0.618 | 0.689 | 0.763 | 0.502 | 0.722 | 0.834 |
| BGS | 0.308 | 0.703 | 0.679 | 0.502 | 0.635 | 0.665 |
| MAML | 0.570 | 0.660 | 0.591 | 0.546 | 0.618 | 0.707 |
| RALE | 0.752 | 0.858 | 0.888 | 0.656 | 0.792 | 0.813 |
| BGSMI | **0.770** | **0.859** | **0.890** | **0.664** | **0.814** | **0.855** |

Table 3: Accuracy with different depths.

| Method | Cora | | | Citeseer | | |
|--------|------|------|------|----------|------|------|
| | 2 layers | 4 layers | 6 layers | 2 layers | 4 layers | 6 layers |
| GCN | 0.631 | 0.683 | 0.693 | 0.603 | 0.636 | 0.643 |
| GAT | 0.624 | 0.689 | 0.670 | 0.703 | 0.733 | 0.711 |
| CHEB | 0.635 | 0.701 | 0.697 | 0.637 | 0.694 | 0.683 |
| GMI | 0.627 | 0.696 | 0.723 | 0.633 | 0.702 | 0.713 |
| BGS | 0.503 | 0.563 | 0.544 | 0.551 | 0.600 | 0.561 |
| MAML | 0.611 | 0.653 | 0.662 | 0.596 | 0.646 | 0.624 |
| RALE | 0.733 | 0.822 | 0.833 | 0.722 | 0.761 | 0.784 |
| BGSMI | **0.746** | **0.887** | **0.847** | **0.738** | **0.772** | **0.793** |

We find that: (a) On Cora, BGSMI achieves the highest accuracy under 1-way 3-shot, 1-way 5-shot, 3-way 5-shot, with the highest average accuracy of 84.0%. On Citeseer, BGSMI also achieves the highest accuracy under 1-way 3-shot, 1-way 5-shot, 3-way 5-shot, with the highest average accuracy of 77.8%. (b) On Cora, BGSMI improves almost 2.4%, 0.1% and 0.2% accuracy under the 1-way 3-shot, 1-way 5-shot, 3-way 5-shot compared with the highest accuracy of other comparison models, respectively. On Citeseer, BGSMI improves almost 1.2%, 2.8% and 5.2% accuracy under the 1-way 3-shot, 1-way 5-shot and 3-way 5-shot compared with the highest accuracy of other comparison models, respectively. These results verify the effectiveness of our BGSMI to improve the accuracy of few-shot learning.

**Alleviation of over-fitting and over-smooth.** To test how MI alleviates over-fitting and over-smooth, we compared the accuracy of few-shot learning based on BGSMI and comparison methods by varying different depths of BGS, reported in Table 3.

We find that: (a) On Cora, the average accuracy of BGSMI keeps almost 82.7% as the highest with the increase of depths of BGS. On Citeseer, BGSMI also achieves the average accuracy of 76.8% with the increase of depths. (b) On Cora and Citeseer, BGSMI improves 1.7% and 1.1% accuracy respectively, compared with other comparison models when the layers increase to 6. (c) On Cora and Citeseer, the average rate of accuracy increase/decrease of BGSMI from 2 layers to 6 layers is 1.1%, while 34.9% of other compari-son methods, which shows the accuracy of BGSMI remains stable with the increase of depths of BGS. This means that the correlation in BGSMI indeed alleviates the over-smooth and over-fitting in few-shot learning.

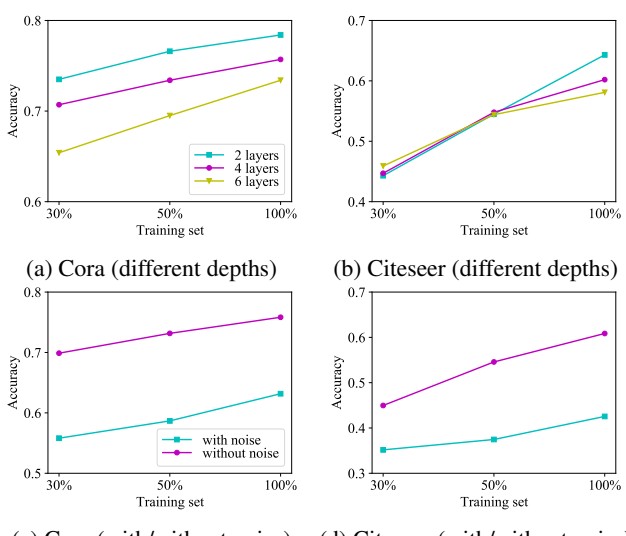

(a) Cora (different depths)   (b) Citeseer (different depths)

(c) Cora (with/without noise)   (d) Citeseer (with/without noise)

Figure 2: Impacts of training size on accuracy of BGSMI.

**Impacts of parameters.** To evaluate the impacts of experimental variables, we recorded the accuracy of BGSMI based few-shot learning with the increase of training size and different parameters, reported in Figure 2. Meanwhile,

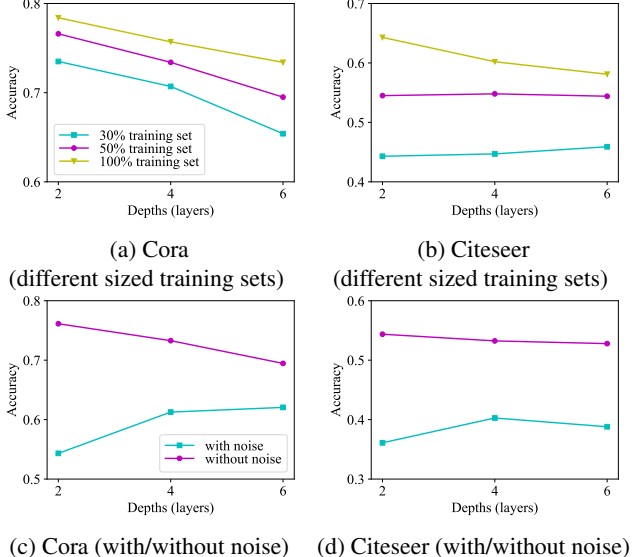

(a) Cora
(different sized training sets)

(b) Citeseer
(different sized training sets)

(c) Cora (with/without noise)

(d) Citeseer (with/without noise)

Figure 3: Impacts of depths on accuracy of BGSMI.

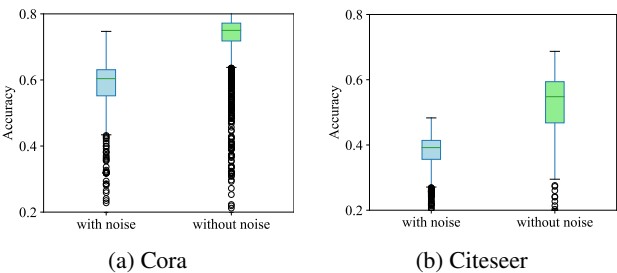

(a) Cora

(b) Citeseer

Figure 4: Impacts of noise on accuracy of BGSMI.

- By incorporating with the MI-based correlation, BGSMI outperforms other state-of-the-art methods. Specifically, BGSMI improves 67.8% and 29.7% average accuracy over BGS on Cora and Citeseer respectively with limited training data. Moreover, BGSMI also achieves the highest accuracy on Cora and Citeseer with different depths of BGS.

- Efficient approximation of MI could be achieved and BGSMI is insensitive to noise. Specifically, the difference between the average accuracy of BGSMI with noise and that without noise is less than 0.2 on different datasets.

# 6 CONCLUSION AND FUTURE WORK

In view of the insufficient features in few training data, we propose a framework BGSMI to leverage the feature correlation described by MI to improve the accuracy of BGNN based few-shot learning. Without proposed framework not only achieves high accuracy, but also alleviates the over-smooth and over-fitting with limited training data in few-shot learning tasks. Experimental results show that the noise will not make BGSMI destroyed.

As further work, we will study how to enhance our framework to eliminate the influence of noise as much as possible. For better interpretability of the combination of deep neural network and BN, we will consider incorporating the ideas of Bayesian deep learning models for further integration of BNCV and BGS.

### Acknowledgements

This work was supported by the National Natural Science Foundation of China (U1802271, 62002311), Program of Key Lab of Intelligent Systems and Computing of Yunnan Province (202205AG070003), Science Foundation for Distinguished Young Scholars of Yunnan Province (2019FJ011) and Major Project of Science and Technology of Yunnan Province (202202AD080001).

we recorded the accuracy with the increase of depths and different parameters, reported in Figure 3.

From Figure 2, we can see that on Cora and Citeseer, the accuracy increases with the increase of training size on both datasets with different depths, and the accuracy does not decrease sharply when adding noise on both datasets with different sized training sets.

From Figure 3, we can see that: (a) high accuracy of few-shot learning on Cora could be kept, and the accuracy on Citeseer remains stable, while the accuracy decreases with the increase of depths of BGS. (b) On Cora and Citeseer, the accuracy remains stable with the increase of depths after adding noise. The above results show that BGSMI is robust to noise to a certain extent, and BGSMI based few-shot learning could achieve high accuracy with limited training data under different depths of BGS.

**Impacts of noise.** MI may intensify the impacts of noise on accuracy, so we evaluated the accuracy of BGSMI based few-shot learning by adding the Gaussian noise, reported in Figure 4. It tells us that, the average accuracy on Cora remains at 60% with noise, which is very close to the minimum accuracy without noise. On Citeseer, the difference between the average accuracy with noise and the quarter-point accuracy without noise is less than 0.1. This shows that the accuracy of BGSMI does not decrease sharply when adding the Gaussian noise.

**Summary.** Following the above experimental results with different experimental variables and datasets, we find that more useful features could be provided by incorporating the MI-based correlation into BGNN and indeed improve the accuracy of BGSMI based few-shot learning.

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
