# OpenReview forum: "Mutual Information Based Bayesian Graph Neural Network for Few-shot Learning"
_auai.org/UAI/2022/Conference — UAI 2022 Oral_

### Official Review · Reviewer_YZuc · 2022-03-22

**Q2(1) Originality/Novelty:** 3
**Q2(2) Significance/Impact:** 3
**Q2(3) Correctness/Technical Quality:** 4
**Q2(6) Clarity Of Writing:** 4
**Q6 Overall Score:** 8
**Q8 Confidence In Your Score:** 4

**Q1 Summary And Contributions:**

This paper proposes a novel method to extract effective features for few-shot learning by incorporating mutual information into Bayesian graph neural network (BGNN). The key contributions include the idea of characterizing the correlation among features in hidden layers of BGNN by adopting MI to describe more useful features, the method of approximating MI by probabilistic inferences over a Bayesian network with continuous variables (BNCV), and the experimental study of the proposed approach.

**Q2 Assessment Of The Paper:**

More detailed information regarding each of these aspects is given below:

**Q2(4) Quality Of Experiments (Optional):**

3: Good: The experimental evaluation is adequate, and the results convincingly support the main claims.

**Q2(5) Reproducibility:**

3: Good: Key resources (e.g., proofs, code, data) are available and key details (e.g., proofs, experimental setup) are sufficiently well-described for competent researchers to confidently reproduce the main results.

**Q3 Main Strengths:**

1.	There are some research findings that attempt to include the mutual information into deep neural networks, but this approach is a novel solution to few-shot learning task.
2.	The idea of using MI to represent the correlation among features is well-motivated and reasonable.
3.	Building BNCV from the Dropout in hidden layers of BGS is interesting and approximating the MI via probabilistic inferences over BNCV is technically correct.
4.	Comprehensive experiments have been conducted and the results show the effectiveness of the approach.


**Q4 Main Weakness:**


1.	The experimental results would be more convincing if more datasets could be adopted.
2.	Some concepts in Section 3 are not properly explained.


**Q5 Detailed Comments To The Authors:**


1.	The proposed BGSMI method significantly improves the accuracy of few-shot learning compared with BGS in Table 2 and Table 3, but are worse than RALE in some situations, which should be discussed in more details.
2.	What is definition of graph G (Section 3)? How to obtain G from the datasets Cora and Citeseer (Section 5.1)? These should be interpreted.
3.	Is there any difference or relationship between D and I in Section 3?
4.	MI is usually used to describe the dependence among data. Is the term dependence more properly than correlation? The motivation of correlation should be further discussed properly.


**Q7 Justification For Your Score:**

The paper proposes a framework BGSMI to leverage the feature correlation described by MI to improve the accuracy of BGNN based few-shot learning. The comprehensive experiments have been conducted and the results show the effectiveness of the approach.

**Q9 Complying With Reviewing Instructions:**

1: Yes.

---

### Official Review · Reviewer_yooL · 2022-03-31

**Q2(1) Originality/Novelty:** 3
**Q2(2) Significance/Impact:** 2
**Q2(3) Correctness/Technical Quality:** 3
**Q2(6) Clarity Of Writing:** 2
**Q6 Overall Score:** 4
**Q8 Confidence In Your Score:** 4

**Q1 Summary And Contributions:**

In this work, the authors proposed a Bayesian graph neural network that adopts the Bayesian network with continuous variables for effective calculation of the mutual information between the different layers of the graph. They approximate the mutual information values efficiently by probabilistic inferences over the Bayesian graph neural network. Authors conduct multiple experiments on two benchmarks to show proposed achieves better performance against existing state-of-the-art methods.

**Q2 Assessment Of The Paper:**

More detailed information regarding each of these aspects is given below:

**Q2(4) Quality Of Experiments (Optional):**

2: Fair: The experimental evaluation is weak: important baselines are missing, or the results do not adequately support the main claims.

**Q2(5) Reproducibility:**

3: Good: Key resources (e.g., proofs, code, data) are available and key details (e.g., proofs, experimental setup) are sufficiently well-described for competent researchers to confidently reproduce the main results.

**Q3 Main Strengths:**

The paper presents an approach that is novel for handling the problem of for Few-Shot Node Classification on graph. The main idea of adopting the Bayesian network with continuous variables to improve the correlation representation among the node features. is a novel idea, at least in this context.

The paper shows that the idea is effective, yielding state-of-the-art results on Few-Shot Node Classification benchmarks, Cora and Citeseer.


**Q4 Main Weakness:**

1. According to the experiment section of the paper, the task concerned in this paper is the Few-Shot Node Classification on graph, but it is described as the “Few-shot learning” in the whole paper, which could be misleading.
2. The results are not so convincing, first, it seems that the method proposed in this paper is not completely superior to previous methods (1-way 3-shot in Citeseer). Second, the paper could lack results on some other datasets, such as Amazon, Email and Reddit, which are mentioned in the previous methods like RALE.
3. The authors do not compare the noise stability of the proposed method with other methods. Although the proposed method is very stable to noise, we do not know how stable it is compared with other methods.


**Q5 Detailed Comments To The Authors:**

First, the authors should reorganize the description of the target task. Second, authors could prove the effectiveness of their strategies by comparing their work with other works on more datasets. Finally, the authors could compare the noise stability of the proposed method with other methods to certificate the noise stability of the proposed method.

**Q7 Justification For Your Score:**

The authors propose a Bayesian graph neural network to solve the Few-Shot Node Classification problem on graph, which is a novel idea. However, since the results are not convincing and the confusing description of the target task, the paper could be misleading.

**Q9 Complying With Reviewing Instructions:**

1: Yes.

---

### Official Review · Reviewer_Eyyy · 2022-04-11

**Q2(1) Originality/Novelty:** 2
**Q2(2) Significance/Impact:** 2
**Q2(3) Correctness/Technical Quality:** 3
**Q2(6) Clarity Of Writing:** 3
**Q6 Overall Score:** 6
**Q8 Confidence In Your Score:** 3

**Q1 Summary And Contributions:**

Propose a framework BGSMI to leverage the feature correlation described by MI to improve the accuracy of BGNN based few-shot learning.

**Q2 Assessment Of The Paper:**

More detailed information regarding each of these aspects is given below:

**Q2(4) Quality Of Experiments (Optional):**

3: Good: The experimental evaluation is adequate, and the results convincingly support the main claims.

**Q2(5) Reproducibility:**

3: Good: Key resources (e.g., proofs, code, data) are available and key details (e.g., proofs, experimental setup) are sufficiently well-described for competent researchers to confidently reproduce the main results.

**Q3 Main Strengths:**

The proposed method is effective for few-shot learning and outperforms some state-of-the-art competitors, and it can handle over-fitting, over-smooth, and noise.

**Q4 Main Weakness:**

It is better to say BNCV than mutual information since it actually incorporates the BNCV method.

**Q5 Detailed Comments To The Authors:**

My main concern is how to actually use the idea of mutual information.  This work is just using BNCV. I feel the title is kind of misleading or a little bit big. Therefore, I wonder if you are thinking of ideas of using MI other than BNCV, even just simply pairwise or something. I bit doubt the effectiveness of upgrading to BNCV (I am not targeting MI). I wish the author need more to justify this. This impacts the significance of this work.

**Q7 Justification For Your Score:**

As I said above, the main problem is the only adoption of BNCV impacts.

**Q9 Complying With Reviewing Instructions:**

1: Yes.

---

### Official Review · Reviewer_1m5N · 2022-04-12

**Q2(1) Originality/Novelty:** 3
**Q2(2) Significance/Impact:** 3
**Q2(3) Correctness/Technical Quality:** 3
**Q2(6) Clarity Of Writing:** 3
**Q6 Overall Score:** 7
**Q8 Confidence In Your Score:** 2

**Q1 Summary And Contributions:**

The authors develop a novel method for using Bayesian graph networks to perform few-shot learning, by using mutual information between the GNN's learned features as a measure of the features' relatedness. On two datasets, their model outperforms all the competitor models.


**Q2 Assessment Of The Paper:**

More detailed information regarding each of these aspects is given below:

**Q2(5) Reproducibility:**

3: Good: Key resources (e.g., proofs, code, data) are available and key details (e.g., proofs, experimental setup) are sufficiently well-described for competent researchers to confidently reproduce the main results.

**Q3 Main Strengths:**

* clear exposition of the derivations, motivations, related work, and experiments
* very interesting idea
* a large number and range of competitor models


**Q4 Main Weakness:**

* the two tested datasets are very similar. It would have been more informative to test on two datasets that are more different


**Q5 Detailed Comments To The Authors:**

* there is a discussion of feature learning in few-shot learning, but I think that there may need to be more of a distinction. Feature learning / representation learning can often be done during meta-training, before few-shot learning occurs?
* seems to be some typos, e.g. "Without proposed framework" in the conclusion


**Q7 Justification For Your Score:**

* It is a very interesting idea and they demonstrate usefulness on at least a narrow set of few-shot learning problems. At the same time, I am not confident because I am not an expert in Bayesian graph neural networks.

**Q9 Complying With Reviewing Instructions:**

1: Yes.

---

### Decision · Program_Chairs · 2022-05-15

**Decision:**

Accept (Oral)

**Comment:**

Meta Review: The paper proposes a new few-shot learning method using mutual information to model relationships among features.

Quality: The paper is technically sound and has a convincing experimental evaluation.

Clarity: The paper is well written.

Originality: The idea is very interesting, and has been demonstrated very useful in few-shot learning.

Significance: the method has great potential in few-shot learning.

Three reviewers are very positive about the paper. The idea is excellent, and the experimental results are convincing.

One reviewer raises some concerns. The authors have responded to the concerns. The authors have provided quite reasonable explanations for the reviewer's concerns.